# How Many Hours of Device Wear Time Are Required to Accurately Measure Physical Activity Post Stroke?

**DOI:** 10.3390/ijerph19031191

**Published:** 2022-01-21

**Authors:** Natalie A. Fini, Anne E. Holland, Julie Bernhardt, Angela T. Burge

**Affiliations:** 1Physiotherapy Department, Alfred Health, Melbourne 3004, Australia; a.holland@alfred.org.au (A.E.H.); angela.burge@monash.edu (A.T.B.); 2Physiotherapy Department, The University of Melbourne, Parkville 3010, Australia; 3Respiratory Research@Alfred, Central Clinical School, Monash University, Melbourne 3004, Australia; 4Stroke Division, Florey Institute of Neurosciences and Mental Health, Heidelberg 3084, Australia; julie.bernhardt@florey.edu.au

**Keywords:** physical activity, stroke, measurement, wear time, activity monitoring

## Abstract

Background. Inadequate physical activity participation is a risk factor for secondary stroke. Before implementing appropriate management strategies, we need to accurately measure the physical activity of stroke survivors. We aimed to determine the duration of physical activity monitoring post-stroke that constitutes a valid day. Methods. We sampled stroke survivors’ physical activity for one week following discharge from inpatient rehabilitation using the Sensewear Armband (Bodymedia, Pittsburgh, PA, USA). To determine the impact of total daily wear time on activity estimate (sedentary, light, and moderate to vigorous physical activity) accuracy, we performed simulations, removing one, two, three, or four hours from a 14-h reference day, and analysed them with linear mixed models. Results. Sixty-nine participants (46 male, 65 ± 15 years) with 271 days of physical activity data were included. All physical activity variables were significantly underestimated for all data sets (10, 11, 12, or 13 h) compared to the 14-h reference data set. The number of days classified as not meeting physical activity recommendations increased as daily monitoring duration decreased: 13% misclassification with 10-h compared to 14-h dataset (*p* = 0.011). Conclusions. The accuracy of physical activity estimates increases with longer daily monitoring periods following stroke, and researchers should aim to monitor post-stroke physical activity for 14 daytime hours.

## 1. Introduction

Participation in physical activity is an important lifestyle-related risk factor for stroke survivors and is essential for the prevention of stroke and cardiovascular disease [1]. Secondary prevention is of paramount importance to stroke survivors [2], yet many studies have documented low physical activity levels following stroke [3]. There are several factors that may account for low post-stroke physical activity, including both physical (e.g., weakness, poor balance, fatigue) and non-physical (e.g., reduced problem solving and motivation) impairments [4].

Globally, many physical activity guidelines recommend that adults undertake 30 min of moderate-intensity physical activity on most days [5,6,7,8]. In the post-stroke clinical setting, it is important to understand whether or not stroke survivors are meeting these physical activity recommendations in order to guide appropriate physical activity management for both the prevention of future chronic disease and to maximise recovery from and participation in life after stroke. Stroke rehabilitation is essential to allow survivors of strokes to reach their full physical recovery to enable physical activity participation [9]. Interventions involving technology are emerging in rehabilitation and show some promising signs of efficacy for physical rehabilitation [10,11,12]. In research settings, it is critical to accurately measure post-stroke physical activity to understand whether physical activity interventions are effective and lead to improvement in physical activity levels and adherence with recommendations, which will ultimately lead to improved health [13]. Therefore, the ability to accurately measure physical activity in stroke survivors is essential.

It is accepted that objective or device measurement of physical activity is more reliable than self-reported measures of physical activity, which are subject to inaccurate recall and over-reporting [14]. To accurately determine habitual physical activity levels following stroke, device wear time or the duration of physical activity monitoring must be adequate. In the stroke literature, accelerometer wear time is highly variable in terms of the number of days of monitoring and the average number of hours physical activity is monitored each day, which ranges from 6–7 h [15] to >23.5 h [16]. Previous studies have examined questions pertaining to how many days and which days (weekend versus weekdays) physical activity should be monitored following stroke, [17,18] but to date, no one has examined how long physical activity should be monitored each day for that day to be considered a valid representation of habitual physical activity.

In healthy populations, the number of hours of accelerometer time required to constitute a valid day has been examined by Herrman et al., 2013 [19], who used the Actigraph (ActiGraph LLC, Pensacola, FL, USA) to measure time inactive and time spent in light, moderate, and vigorous activity. The authors used 14 h as a reference and created semi-simulation datasets of 10, 11, 12, and 13 h and then compared the absolute per cent error of each with the 14-h day reference [19]. They concluded that 13 h per day of physical activity measurement is required to constitute a valid day [19]. Another paper that investigated this issue in 100 healthy participants with the Actigraph used a random data removal technique to create simulated data sets from a 15-h reference day [20]. They established that daily estimates of physical activity improved with increasing wear time and that with lower wear time, participants were misclassified as meeting or not meeting physical activity guidelines more than 40% of the time. Conversely, in a large study of obese adults with Type II diabetes, no statistically significant differences in the number, intensity, or duration of bouts of moderate or vigorous physical activity were found between different daily durations of measurement (8, 10, or 12 h of wear time) [21]. In this study, however, participants performed very low levels of physical activity, which is likely to have influenced the results as minimal bouts of moderate to vigorous physical activity were observed (on average less than one 10-min bout per day). It appears that in healthy participants, a higher device wear time produces more a more accurate reflection of habitual physical activity, but whether this applies to clinical populations is unclear.

This study’s primary aim was to determine how many hours of device wear time are required to accurately measure physical activity post-stroke. We were specifically interested in the impact of daily duration of physical activity monitoring on:(a)measures of daytime physical activity, specifically sedentary awake time, light and moderate to vigorous physical activity(b)the attainment of guideline-recommended physical activity levels in stroke survivors

## 2. Materials and Methods

The data for this study were taken from the baseline measurements of a prospective longitudinal study of primary stroke survivors [16]. The study included patients with a diagnosis of primary stroke admitted to a neurological rehabilitation unit in a metropolitan hospital who were invited to participate in the longitudinal study. Exclusion criteria were minimal: palliative diagnosis, living more than two hours from the hospital, and rehabilitation admission of less than five days. Ethics approval was obtained from the local hospital and university committees.

Participants completed a number of measures in their baseline assessment, including measures of mobility (e.g., gait speed, six-minute walk test), cardiovascular risk (e.g., blood pressure, body mass index (BMI)), cognitive function (Montreal Cognitive Assessment), fatigue (Fatigue Severity Scale), and mood (Hospital Anxiety and Depression Scale) [22]. At the end of the assessment, the participants were fitted with a triaxial accelerometer (Sensewear Armband, Bodymedia, Pittsburgh, PA, USA) to monitor their physical activity for the following week.

The Sensewear Armband (Bodymedia, Pittsburgh, PA, USA) was worn over the triceps of the unaffected arm at all times except for water-based activities (e.g., showering) for a period of one week. The device comprises a triaxial accelerometer with sensors that measure skin temperature, heat flux, and galvanic skin response to then report measures of intensity (e.g., energy expenditure in joules and average metabolic equivalents (METS)), frequency (e.g., step count), and duration (e.g., time spent physically active, time spent sleeping). The Sensewear Armband (Bodymedia, Pittsburgh, PA, USA) has been shown to be valid for measuring energy expenditure in the chronic phase post-stroke [23,24], and it has been shown to be reliable for measuring physical activity following stroke [25].

### Data Analysis

The data were analysed using SPSS version 26.0 (IBM, New York, NY, USA). The results used to describe participants are presented as means and standard deviations for normally distributed data and medians and inter-quartile ranges for data that were not normally distributed.

The Sensewear version 7.0 software (Bodymedia, Pittsburgh, PA, USA), which uses a proprietary-limited algorithm, was used to process the raw data from the Sensewear Armband (Bodymedia, Pittsburgh, PA, USA) for each participant. This algorithm provides data coded each minute as spent sleeping, sedentary (awake), or in light, moderate, or vigorous physical activity. We set the parameters for those as follows: 0–1.5 METS sedentary; 1.6–3.0 METS light-intensity physical activity (LIPA), and>3.0 METS moderate to vigorous physical activity (MVPA) [5]. The MET provides information about how much the body is working, where one MET is the amount of energy used during quiet sitting [5]. Data from the Sensewear Armband (Bodymedia, Pittsburgh, PA, USA) were exported to the Microsoft Excel (IBM, New York, NY, USA) program.

Specific inclusion criteria for this analysis were a minimum accelerometer wear time of five days with at least 10 daytime hours (between 7 a.m. and 9 p.m.) each day. From there, only days with at least 11 valid hours of continuously collected data were included in this analysis. For an hour to be considered valid, at least 40 min of the hour needed to be captured [19]. For each day of accelerometer data, hours prior to 7 a.m. and after 9 p.m. were removed, as were any periods of non-wear (more than 20 min per hour) [19]. If at least 11 continuous hours remained, that day was included. Therefore, days of 11, 12, 13, and 14 h were included. The methodology for data preparation used in McGrath et al., [20] was replicated as follows. For each day, a range of simulations were created. A random number generator (https://www.calculator.net/random-number-generator.html, accessed on 23 March 2021)) was used to randomly select one, two, three, or four hours (60-min segments) to be removed, to create simulated days with 13, 12, 11, and 10-h days from the original 14-h reference data set. Figure 1 outlines the data simulation. A power calculation for sample size was not applied as this is a secondary analysis of data.

Daily estimates of time spent in each of sleep, sedentary (awake), LIPA, and MVPA were calculated for each day and each simulation for each day. Differences in daily estimates for sedentary awake time, time in LIPA, and time in MVPA between the 14-h reference set and the simulated data sets (10, 11, 12, and 13 h) were analysed using linear mixed models. Model covariates (age, gait speed, and BMI) were determined a priori. These covariates were chosen as they have been shown to be consistently associated with physical activity levels [26,27]. Model assumptions were evaluated, and the most appropriate data transformations were examined where required [28]. Linear mixed models have been acknowledged as suitable for repeated measures analyses [29]. Absolute per cent error (APE) was calculated for each time estimate in the simulated data sets using the 14-h reference set ([(simulated − reference value)/reference value] × 100) where smaller values represent less difference between the simulated and reference data sets [19,20].

To determine if the monitoring duration affected whether or not participants were meeting recommendations of 30 min of MVPA per day, Cochran’s Q test was applied to days with 14 h of data collected. Days were classified as having 30 min of MVPA recorded or not recorded at each of the 14-h reference data set and 13, 12, 11, and 10-h simulated days.

## 3. Results

Sixty-nine participants met the inclusion criteria. The total number of days included in the analysis was 271:229 days with 14 h of data; 238 days with 13 h of data; 254 days with 12 h of data; 271 days with 11 h of data.; and 271 days with 10 h of data. See Table 1 for details of included participants.

As the time estimates did not meet model assumptions, model parameters were optimised using square root transformation (time in LIPA, time in MVPA) and inverse square root transformation (sedentary awake time) [28]. However, as these results are unable to be back-transformed, [30] the results for the raw data are presented here for ease of interpretation of the magnitude of difference for time estimates for each physical activity outcome within each simulated data set (relative to 14-h reference set) (Table 2) [31]. The results for the transformed data are presented in the online supplement (Appendix A Appendix A). Both approaches revealed statistically significant underestimation of each time estimate for all simulated data sets for each physical activity outcome (relative to 14-h reference set), with a similar pattern observed.

There was a significant difference in the proportion of days on which 30 min of MVPA was recorded between reference and simulated data sets (Cochran’s Q test *p* = 0.011, Table 3). Fewer hours of data resulted in more days where a minimum of 30 min of MVPA was not recorded, with 13% of participants being misclassified with 10 h of monitoring compared to the 14-h reference dataset.

## 4. Discussion

This paper demonstrates that following stroke, more accurate estimates of physical activity are obtained with longer daily monitoring periods. Between 10 and 14 h of daily measurement, physical activity estimates for each variable—sedentary awake time, time in LIPA, and time in MVPA—significantly improved, and the absolute per cent error decreased with each increasing hour of measurement. Another important finding was that wear time significantly impacted whether a day was classified as having met or not met 30 min of MVPA. This misclassification could impact the overall management of stroke survivors in terms of life-style-related risk factors for the prevention of future stroke and cardiovascular disease.

The results of this study highlight the need for an adequate monitoring duration to gain an accurate picture of habitual physical activity following stroke. This was relevant across each of the physical activity variables. Even missing only 1 h resulted in a mean difference of 5 min less (95% CI between 1 and 10 min less) MVPA recorded. Interestingly, for every hour of missed data, the absolute per cent error increased by approximately 7% for each of sedentary time, LIPA, and MVPA. This phenomenon was also seen in the analysis by McGrath et al., 2017 [20], in their similar analysis of physical activity in healthy people—for every less hour of data collected between 15 and 10 h, the absolute per cent error increased by 5–7% for all variables. Likewise, for sedentary behaviour and LIPA measured in healthy populations, Herrmann et al., 2013 [19], observed an increase of 7% in absolute per cent error for each hour less of monitoring time. They measured moderate and vigorous physical activity separately, which makes it difficult to compare those findings.

It is important not to place unwarranted burden on stroke survivors by asking them to wear accelerometers for longer than necessary; however, if participants go to the trouble of wearing an accelerometer in the first instance, it is critical that the information obtained is useful and accurate. Previous research has shown that a two-day wear period is sufficient for simple measures of physical activity, such as step count and time in moderate to vigorous physical activity, and three or more days is required for more complex measures, such as bouts of activity and inactivity [18]. However, it is important to also be aware of how much wear-time in a given day will ensure that that day provides accurate data and contributes to the days of wear-time required. As wearable devices are becoming more commonplace and comfortable, and unobtrusive to wear, 24-h activity monitoring protocols may become more realistic. Many studies of physical activity after stroke only monitor activity for 8–12 h of the day, [15,32,33,34] and as such, may be underestimating the amount of sedentary behaviour and light, moderate, and vigorous physical activity is undertaken. Looking at the modelling in this study, measuring physical activity for 12 h of the day could underestimate time spent in each category by 14% and measuring for as little as 8 h per day could underestimate time spent in each category by up to 42%, which is quite considerable. This may be one reason why physical activity has been found to be so low in many observational studies post stroke. Interestingly, for the participants in this study, where physical activity was measured for more than 23.5 h per day on average, the median time spent in MVPA per day was 62 min, indicating that physical activity guidelines were being met.

Findings from the present study indicate that shorter wear times result in underestimation of physical activity in each category, and as such, this will lead to misclassification of whether or not stroke survivors meet physical activity guidelines. This may impact the physical activity interventions and overall management provided to stroke survivors for future cardiovascular disease prevention. It may lead to stroke survivors being informed that they are not meeting their goals, which may be a motivating factor for some; however, for many, it could lead to disappointment, dissatisfaction, and a decrease in motivation [35]. A worst-case scenario would be subsequent discontinuation of their physical activity program and an increase in cardiovascular risk. This again emphasises the importance of accurate physical activity measurement.

With the continuing development of device-based technologies, longer monitoring periods (e.g., 24 h or devices worn for the entire day) may become the norm. Telerehabilitation using digital communication technology is an approach that allows safe remote monitoring of physical activity post-stroke by clinicians [36]. Clinicians could set targets for their patients and remotely review progress via a device app or platform and provide feedback and review goals with the patient via telephone or video conferencing without the need for face-to-face consultation. This is one approach that is being used widely across many health conditions, particularly since the beginning of the COVID-19 pandemic [37].

### Limitations

There are several limitations to this study. Firstly, while the Sensewear Armband (Bodymedia, Pittsburgh, PA, USA) has been shown to be valid for measuring physical activity in chronic stroke, in the acute and sub-acute phases, its validity for measuring energy expenditure is lower [38,39]. Another limitation is that we only included “daytime hours” between 7 a.m. and 9 p.m. in our data analysis. This is based on the assumption that modifiable physical activity generally does not occur between 9 p.m. and 7 a.m., and we acknowledge that while this would certainly be the norm, it may not always be the case. Additionally, our simulation technique randomly removed 60-min segments from the reference dataset, and this may not replicate a typical scenario of daily accelerometer wear time, noting it is difficult to predict what a “typical” scenario may be. Finally, the sample of stroke survivors in this study were relatively able and active, which may not be representative of a more severely affected population.

## 5. Conclusions

This paper demonstrates that the accuracy of physical activity measurement will improve with a longer monitoring period. Researchers should aim to obtain 14 h of daytime monitoring when assessing physical activity after stroke. This will enable accurate classification of whether stroke survivors are meeting physical activity guidelines and facilitate targeted life-style related risk factor management.

## Figures and Tables

**Figure 1 ijerph-19-01191-f001:**
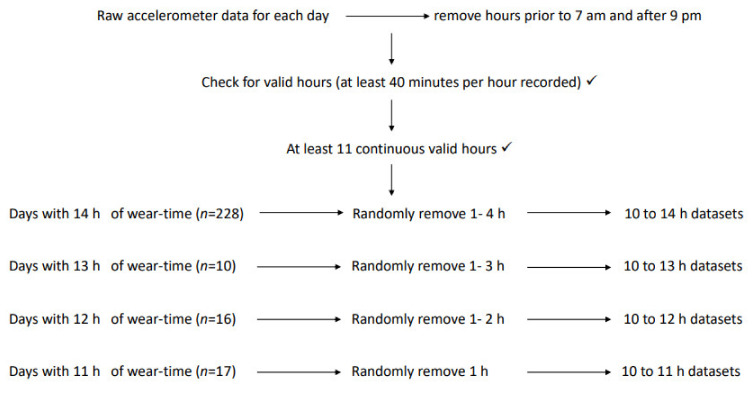
Data preparation and simulation summary. ✓ = criterion satisfied; h = hours.

**Table 1 ijerph-19-01191-t001:** Participant characteristics.

Characteristics	*n* = 69
Gender, *n* male (%)	46 (67)
Age (years), mean (SD)	65 (15)
Type of stroke (infarct/haemorrhage), *n* infarct (%)	48 (70)
NIHSS, mean (IQR)	9 (4, 12)
Time post-stroke (days), median (IQR)	151 (99, 225)
Body mass index (kg/m^2^), mean (SD)	27.0 (5.2)
Gait speed (m/s), median (IQR)	1.2 (0.9, 1.4)
6-min walk test (m), mean (SD)	389 (181)

*n*, number; SD, standard deviation; NIHSS, National Institutes of Health Stroke Scale; IQR, interquartile range.

**Table 2 ijerph-19-01191-t002:** Daily physical activity estimates for 10–14 h datasets.

Original Data	Sedentary Awake Time		Time in LIPA		Time in MVPA	
14-h Reference Set	519 (497 to 542)		154 (138 to 170)		75 (61 to 89)	
	MD (95% CI)	APE	MD (95% CI)	APE	MD (95% CI)	APE
Δ 13	−35 (−45 to −25)	7%	−12 (−19 to −5)	8%	−5 (−1 to −10)	7%
Δ 12	−69 (−80 to −59)	13%	−23 (−30 to −17)	15%	−11 (−17 to −4)	14%
Δ 11	−108 (−118 to −98)	21%	−34 (−41 to −28)	22%	−15 (−22 to −9)	20%
Δ 10	−145 (−155 to −135)	28%	−45 (−52 to −38)	29%	−20 (−27 to −14)	27%

Results are minutes per day unless indicated. APE, absolute per cent error; LIPA, light-intensity physical activity; MVPA, moderate to vigorous-intensity physical activity.

**Table 3 ijerph-19-01191-t003:** Proportion of days that 30 min of MVPA is/is not recorded.

	Did Not Record 30 min MVPA	Did Record 30 min MVPA	
14 h	82	147	Cochran’s Q
13 h	90	139	*p* = 0.011
12 h	99	130	
11 h	107	122	
10 h	112	117	

MVPA, moderate to vigorous-intensity physical activity.

## Data Availability

The data presented in this study are available on request from the corresponding author. The data are not publicly available due to ethical restrictions.

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
