# Peer review of "How Many Hours of Device Wear Time Are Required to Accurately Measure Physical Activity Post Stroke?"

_ijerph, 2022, doi:10.3390/ijerph19031191_

Round 1

Reviewer 1 Report

The manuscript is suitable for publication, it leaves the reader with those constructive doubts on the subject by expanding the discussion in the literature... leaving only minimal suggestions.

Thanks

Line 97 a Photo or Figure?

Line 190 I suggest adding a statement as follows “Moreover, Telerehabilitation might an approach that, using digital communication technology, allows both assessment and remote monitoring of patients during exercise efficiently and safely”
(ref: https://doi.org/10.1108/JET-11-2020-0047 )

Author Response

Reviewer #1

The manuscript is suitable for publication, it leaves the reader with those constructive doubts on the subject by expanding the discussion in the literature... leaving only minimal suggestions.

Response:

Thank you for your helpful review of our study.

  1. Line 97 a Photo or Figure

Response:

We can add a photograph of the Sensewear Armband if the editors deem it appropriate. We have attached a photograph of the Sensewear Armband in the submission. Please note that if included, the formatting of the document will need to be changed as will the original Figure in the article – it will need to be changed to Figure 2, as will reference to it in the text.

  1. Line 190 I suggest adding a statement as follows “Moreover, Telerehabilitation might an approach that, using digital communication technology, allows both assessment and remote monitoring of patients during exercise efficiently and safely”
    (ref: https://doi.org/10.1108/JET-11-2020-0047 )

Response:

Thank you for your suggestion. As per reviewer 3 we have added a paragraph near the end of the discussion about future developments and have referred to telerehabilitation there as follows:

“With the continuing development of device-based technologies longer monitoring periods (e.g. 24 hours or devices worn for the entire day) may become the norm. Telerehabilitation using digital communication technology is an approach that that allows safe remote monitoring of physical activity post-stroke by clinicians.36 Clinicians could set targets for their patients and remotely review progress via a device app or platform and provide feedback and review goals with the patient via telephone or videoconferencing without the need for face-to-face consultation. This is one approach that is being used widely across many health conditions, particularly since the beginning of the COVID-19 pandemic.37

Reviewer 2 Report

This is an interesting, important, and well-written manuscript. The methods are described adequately and in detail. The results of the study will be of interest to the readers, clinicians, and researchers working in this area.   

Author Response

Thank you for your kind review of our study.

Reviewer 3 Report

In this paper, the authors aim to determine how many hours of device wear time are 76 required to accurately measure physical activity post stroke. The paper is easy to read and insightful.

Abstract

In the abstract, +/- should be replaced with ±.

Introduction

In the introduction I would add only a few references to underline the importance of the rehabilitation program in the post-stroke patients:

Maranesi, G.R. Riccardi, V. Di Donna, M. Di Rosa, P. Fabbietti, R. Luzi, L. Pranno, F. Lattanzio, R. Bevilacqua “EFFECTIVENESS OF INTERVENTION BASED ON END-EFFECTOR GAIT TRAINER IN OLDER PATIENTS WITH STROKE: A SYSTEMATIC REVIEW”, Journal of the American Medical Directors Association, 2019, S1525-8610(19)30750-9. DOI: 10.1016/j.jamda.2019.10.010

Maranesi E, Bevilacqua R, Di Rosa M, Pelliccioni G, Di Donna V, Luzi R, Morettini M, Sbrollini A, Casoni E, Rinaldi N, Baldoni R, Lattanzio F, Burattini L, Riccardi GR. AN INNOVATIVE TRAINING BASED ON ROBOTICS FOR OLDER PEOPLE WITH SUBACUTE STROKE: STUDY PROTOCOL FOR A RANDOMIZED CONTROLLED TRIAL. Trials. 2021 Jun 14;22(1):400. doi: 10.1186/s13063-021-05357-8

Materials and Methods

Although the session is well structured, I have a few comments:

  • Only the exclusion criteria have been indicated. I suggest to insert also the inclusion criteria, to understand which kind of patients we are referring to, specifying also the evaluation scales administered and the cut-off values considered.
  • It seems to me that the METS measurement is very important. I suggest to detail it better, specifying how it is calculated
  • In Figure 1, hr should be replaced with hrs in the last column.

Discussion

This session is well structured. I suggest you add a paragraph about possible future developments

Author Response

Reviewer #3

In this paper, the authors aim to determine how many hours of device wear time are required to accurately measure physical activity post stroke. The paper is easy to read and insightful.

Response:

Thank you for your insightful review of our study.

Abstract

  1. In the abstract, +/- should be replaced with ±.

Response:

This change has been made.

Introduction

  1. In the introduction I would add only a few references to underline the importance of the rehabilitation program in the post-stroke patients:

Maranesi, G.R. Riccardi, V. Di Donna, M. Di Rosa, P. Fabbietti, R. Luzi, L. Pranno, F. Lattanzio, R. Bevilacqua “EFFECTIVENESS OF INTERVENTION BASED ON END-EFFECTOR GAIT TRAINER IN OLDER PATIENTS WITH STROKE: A SYSTEMATIC REVIEW”, Journal of the American Medical Directors Association, 2019, S1525-8610(19)30750-9. DOI: 10.1016/j.jamda.2019.10.010

Maranesi E, Bevilacqua R, Di Rosa M, Pelliccioni G, Di Donna V, Luzi R, Morettini M, Sbrollini A, Casoni E, Rinaldi N, Baldoni R, Lattanzio F, Burattini L, Riccardi GR. AN INNOVATIVE TRAINING BASED ON ROBOTICS FOR OLDER PEOPLE WITH SUBACUTE STROKE: STUDY PROTOCOL FOR A RANDOMIZED CONTROLLED TRIAL. Trials. 2021 Jun 14;22(1):400. doi: 10.1186/s13063-021-05357-8

Response:

Thank you for your suggestion, we have added a couple of sentences about the importance of rehabilitation and the emerging use of technologies as follows:

“Stroke rehabilitation is essential to allow survivors of stroke to reach their full physical recovery to enable physical activity participation.9 Interventions involving technology are emerging in rehabilitation and show some promising signs of efficacy for physical rehabilitation.10-12

Materials and Methods

Although the session is well structured, I have a few comments:

  1. Only the exclusion criteria have been indicated. I suggest to insert also the inclusion criteria, to understand which kind of patients we are referring to, specifying also the evaluation scales administered and the cut-off values considered.

Response:

The inclusion criteria for the study were very broad and simply: first ever stroke and admission to rehabilitation at the large metropolitan hospital we recruited from. This has been made more clear in the text by specifying “The study included patients” at the start of the sentence outlining this.

4. It seems to me that the METS measurement is very important. I suggest to detail it better, specifying how it is calculated

Response:

Thank you for this suggestion. The detail around the measurement of METS has been added as follows:

“The MET provides information about how much the body is working, where one MET is the amount of energy used during quiet sitting.5”

5. In Figure 1, hr should be replaced with hrs in the last column.

We disagree with this comment and believe that the grammar is correct in this instance as it is referring to for example 10-14 hour datasets as opposed to days with 14 hours of wear-time.

Discussion

  1. This session is well structured. I suggest you add a paragraph about possible future developments.

This is a great suggestion, thank you very much. We have added a paragraph as you suggest as follows:

“With the continuing development of device-based technologies longer monitoring periods (e.g. 24 hours or devices worn for the entire day) may become the norm. Telerehabilitation using digital communication technology is an approach that that allows safe remote monitoring of physical activity post-stroke by clinicians.36 Clinicians could set targets for their patients and remotely review progress via a device app or platform and provide feedback and review goals with the patient via telephone or videoconferencing without the need for face-to-face consultation. This is one approach that is being used widely across many health conditions, particularly since the beginning of the COVID-19 pandemic.37